# BiLSTM-MLAM: A Multi-Scale Time Series Prediction Model for Sensor Data Based on Bi-LSTM and Local Attention Mechanisms

**DOI:** 10.3390/s24123962

**Published:** 2024-06-19

**Authors:** Yongxin Fan, Qian Tang, Yangming Guo, Yifei Wei

**Affiliations:** 1School of Computer Science, Northwestern Polytechnical University, Xi’an 710072, China; 18810895272@163.com; 2School of Electronic Engineering, Beijing University of Posts and Telecommunications, Beijing 100876, China; tangqian@bupt.edu.cn (Q.T.); weiyifei@bupt.edu.cn (Y.W.); 3School of Cybersecurity, Northwestern Polytechnical University, Xi’an 710072, China

**Keywords:** bidirectional long short-term memory, local attention mechanism, multi-scale patch segmentation, time series prediction

## Abstract

This paper introduces BiLSTM-MLAM, a novel multi-scale time series prediction model. Initially, the approach utilizes bidirectional long short-term memory to capture information from both forward and backward directions in time series data. Subsequently, a multi-scale patch segmentation module generates various long sequences composed of equal-length segments, enabling the model to capture data patterns across multiple time scales by adjusting segment lengths. Finally, the local attention mechanism enhances feature extraction by accurately identifying and weighting important time segments, thereby strengthening the model’s understanding of the local features of the time series, followed by feature fusion. The model demonstrates outstanding performance in time series prediction tasks by effectively capturing sequence information across various time scales. Experimental validation illustrates the superior performance of BiLSTM-MLAM compared to six baseline methods across multiple datasets. When predicting the remaining life of aircraft engines, BiLSTM-MLAM outperforms the best baseline model by 6.66% in RMSE and 11.50% in MAE. In the LTE dataset, it achieves RMSE improvements of 12.77% and MAE enhancements of 3.06%, while in the load dataset, it demonstrates RMSE enhancements of 17.96% and MAE improvements of 30.39%. Additionally, ablation experiments confirm the positive impact of each module on prediction accuracy. Through segment length parameter tuning experiments, combining different segment lengths has resulted in lower prediction errors, affirming the effectiveness of the multi-scale fusion strategy in enhancing prediction accuracy by integrating information from multiple time scales.

## 1. Introduction

With the rapid advancement of the Internet of Things (IoT) and 5G technologies, sensor networks have found widespread applications across various domains, including industrial production, environmental monitoring, intelligent transportation, and energy management. Sensor networks play a crucial role by continuously monitoring and transmitting diverse data such as environmental parameters, device statuses, and user behaviors in real time, thus providing a reliable data foundation for numerous applications. Leveraging the collected data for prediction purposes has become increasingly crucial in this context, facilitating efficient decision-making and management strategies across different fields. For instance, with the rapid digital transformation and the expanding user base of the internet, effective management and prediction of network traffic have gained significant importance. According to the latest report from the China Internet Network Information Center (CNNIC) [1], as of June 2023, the number of internet users in China surged to 1.079 billion, with an internet penetration rate of 76.4%. Internet traffic witnessed substantial growth, reaching a total of 1423 billion GB, marking a 14.6% year-on-year increase. Predicting network traffic can aid in optimizing network resource allocation and enhancing operational efficiency and user experience.

The rapid development of artificial intelligence (AI) technologies offers novel methods and tools to improve the performance of time series prediction in sensor networks [2]. Several studies [3,4,5] have modeled it as a general time series prediction problem and focused on improving prediction performance. The variability of time series data is driven by various factors, including seasonal variations, long-term trends, periodic fluctuations, and random events. Seasonal factors encompass natural seasons, holidays, and differences between working days and weekends, while trend factors involve technological advancements, population growth, and economic development. Additionally, cyclical factors stem from economic cycles and industry-specific fluctuations, while random factors, such as natural disasters and equipment failures, introduce uncertainty and anomalies. Policy changes, management decisions, technological advancements, and societal behaviors also significantly influence data patterns. These factors interact, resulting in data with high dimensionality, nonlinearity, and non-stationarity, making traditional linear prediction models inadequate. Furthermore, the spatial and temporal correlations between data from different sensor nodes further complicate data analysis. For instance, in smart grids, power load data from different regions influence each other, while in intelligent transportation systems, traffic flow in adjacent sections is interrelated.

Addressing these challenges requires researchers to delve into complex data analysis techniques, particularly leveraging the latest advancements in machine learning and deep learning domains. Machine learning, characterized by computational methods that enhance performance or make accurate predictions based on experiences [6], can effectively identify latent patterns and trends from extensive time series datasets, significantly improving prediction accuracy. However, despite the remarkable performance of emerging methods in certain sequence prediction challenges, their effectiveness may not surpass that of traditional methods in specific scenarios. Therefore, to overcome the limitations of traditional models and address the shortcomings of emerging methods, this study aims to conduct an in-depth investigation into sensor network data time series prediction techniques, aiming to achieve higher accuracy and reliability across various application environments, thereby enhancing management and decision-making capabilities in different fields.

The primary contributions of this study include the following:We propose a multi-scale feature fusion prediction method named BiLSTM-MLAM for time series prediction. Experimental validation on multiple datasets demonstrates its outstanding performance.We utilize the Bi-LSTM structure to enable the model to consider past and future contextual information at each time step. Compared to traditional LSTM, Bi-LSTM comprehensively utilizes information in the sequence, aiding in capturing long-term dependencies.We introduce a multi-scale feature fusion mechanism, design the multi-scale patch segmentation module, and obtain long sequences composed of equal-length segments representing multiple time scales by setting different segment lengths. This facilitates the capture of patterns and features at different temporal resolutions.We introduce a local attention mechanism for effective modeling of temporal dependencies within multiple sub-sequence time series, thereby enhancing feature extraction. The local attention mechanism allows explicit extraction of local features from the temporal relationships of sub-sequence time series within long sequences.

## 2. Related Works

Time series prediction is widely applied across various fields. In finance [7], it aids in predicting stock prices, currency exchange rates, and interest rates, empowering investors to make more informed decisions. In meteorology [8], forecasting weather changes, including temperature and rainfall, offer critical insights for daily travel, agricultural production, and emergency management. Traffic prediction [9] supports urban planning and traffic management, while in the medical field [10], it contributes to disease prevention and public health management. In the energy sector [11], forecasting energy demand and prices assists in effective energy planning and management for enterprises and dispatch departments. In practical applications, time series generated by complex systems often exhibit non-stationarity and non-linear characteristics, encompassing both deterministic and stochastic components. Previous research has predominantly relied on mathematical-statistical models, such as autoregressive integrated moving average [12], vector autoregression [13], and generalized autoregressive conditional heteroskedasticity [14]. However, these models, constrained by fixed mathematical formulas, struggle to adequately express the complex features of time series, posing challenges in accurately predicting them. While classical methods like ARIMA models [15,16] and exponential smoothing [17,18] have been utilized in many studies with certain achievements, they often focus on a single time series, overlooking the correlations between sequences and encountering limitations in handling complex time patterns, such as long-term dependencies and non-linear relationships.

Propelled by rapid advancements in machine learning and deep learning, time series prediction algorithms leverage the potential of these cutting-edge technologies, demonstrating remarkable performance. Machine learning methods transform time series problems into supervised learning, utilizing feature engineering and advanced algorithms for prediction, effectively addressing complex time series data. As machine learning applications in research expand, notable models like the random forest model, support vector regression model, and the Bayesian network model have emerged. While these methods excel with straightforward datasets, they face challenges in capturing intricate non-linear relationships among multiple variables with extensive datasets, leading to suboptimal performance.

In recent years, researchers have increasingly relied on artificial neural networks (ANNs) to tackle complex time series prediction problems, owing to their abilities for self-learning, self-organization, adaptability, and robust approximation of non-linear functions. The development of prediction algorithms based on deep neural networks indicates a growing trend [19,20,21,22,23,24,25,26]. Among these, a multi-layer perceptron (MLP) [27] and extreme learning machine (ELM) [28] are frequently utilized in time series prediction. Indrastanti et al. [29] employed multi-layer perceptron to develop a precise flood prediction system. Sven F. Crone [30] attained second place in a synthetic time series competition and the ESTSP 2008 dataset by employing MLP. Alexander Grigorievskiy et al. [31] utilized an optimally pruned extreme learning machine (OP-ELM) for long-term time series prediction. Min Han et al. [32] introduced an innovative method combining a hybrid variable selection algorithm with an enhanced extreme learning machine for predicting multivariate chaotic time series. In domains such as renewable energy and the power market, MLP and ELM demonstrate superior performance compared to traditional statistical approaches and machine learning methodologies, effectively processing intricate energy data, power demand, and market behavior to provide accurate predictions and decision support [33,34,35,36]. However, due to their simplistic structure, ANNs have specific limitations in time feature extraction. Consequently, many studies have shifted to recurrent neural networks (RNNs). Among them, recurrent neural network models, specifically crafted for addressing time series problems, employ gated unit structures to manage information. These models pass information layer by layer through concatenation, recursively updating themselves for predictions, and finding widespread applications in solving prediction problems [24,37,38,39,40,41,42,43]. Among various variants of RNN, long short-term memory (LSTM) has become the most popular model for addressing the exploding and vanishing gradient problems during RNN training [44]. For example, Vinayakumar et al. [45] successfully employed LSTM in backbone networks for traffic prediction. Despite the impressive performance of LSTM in many aspects, it has some limitations, especially in dealing with very long sequences, where a lack of long-term dependencies may arise. As sequences lengthen, LSTM’s memory units may lose vital information, compromising its predictive accuracy. Furthermore, longer sequences exacerbate LSTM’s computational complexity, placing a burden on computational resources and constraining its applicability for handling extended sequences.

In 2017, the Transformer architecture proposed by Google gradually found applications in time series prediction [46]. The Transformer model handles time series data by introducing self-attention mechanisms, effectively capturing repetitive patterns of long-term dependencies. The self-attention mechanism enables the model to assign greater attention weights to important information at different positions in the sequence, thereby enhancing modeling capabilities. Architectures based on attention mechanisms have shown outstanding performance in time series prediction tasks [21,22]. Bryan Lim et al. [47] introduce the temporal fusion transformer (TFT)—a novel attention-based architecture that combines high-performance multi-horizon forecasting with interpretable insights into temporal dynamics. Over time, efficient mixed time series prediction models have started to emerge. For instance, the encoder–decoder model, comprising two LSTMs acting as an encoder and decoder, can proficiently extract features and produce accurate predictions. Liang et al. [42] employed encoder–decoder architecture to investigate two environmental quality datasets in the experiment, demonstrating the satisfactory performance of the method. Recently, time series prediction models that combine convolutional neural networks (CNNs) and graph neural networks (GNNs) have made significant progress. These advanced models leverage CNNs’ strengths in capturing local patterns and GNNs’ abilities to handle relational data structures, offering superior performance for complex time series data. Zhao et al. [48] demonstrated that STGCN-HO, which integrates graph convolution and gated linear units, significantly improves cellular traffic prediction accuracy compared to existing RNN and CNN-grid methods.

Although significant progress has been made in time series prediction methods, there are still some challenges. One major issue is the difficulty in capturing long-term dependencies and complex patterns in non-stationary time series data, which often exhibit sudden changes and trends that traditional models struggle to adapt to. Additionally, existing models may lack the robustness required to handle the inherent noise and randomness in real-world data, leading to suboptimal predictions. Models like LSTM, while proficient at handling certain sequence dependencies, may encounter limitations when dealing with very long sequences, posing problems for maintaining long-term memory. Therefore, the demand for innovative approaches, such as the self-attention mechanisms in Transformer models, becomes crucial. These models offer enhanced capabilities in recognizing and modeling complex patterns within time series data, emphasizing the urgency and necessity for continued advancements in this field to achieve more accurate and reliable predictions.

## 3. Methods

### 3.1. Overall Framework

In predicting future time series based on known time series data, this paper introduces a novel time series prediction model. This model framework, which is termed the multi-scale time series prediction model and is based on Bi-LSTM and local attention mechanism (BiLSTM-MLAM), incorporates three essential modules: bidirectional long short-term memory (Bi-LSTM), multi-scale segmentation, and local attention mechanism (LAM). The schematic diagram illustrating the BiLSTM-MLAM framework is presented in Figure 1.

Assuming an input raw sequence $I=\left\{x_{1},x_{2},x_{3},\ldots ,x_{t}\right\}$ is used to predict $y_{t+1}$, where $x_{i}\in R^{m}$ and *m* represent the feature dimensions. Initially, we input the raw sequence *I* in chronological order into a bidirectional long short-term memory network. Bi-LSTM extracts information in both forward and backward directions, enabling the model to thoroughly consider the current input at each time step, along with contextual information from the past and future. This approach enhances the model’s ability to capture long-term dependencies, learning patterns, and features within the sequence.

Simultaneously, to enable the model to capture and extract features at different time scales, enhancing its perception of various scale features within the time series, we introduce the multi-scale patch segmentation module. This module, based on a set hyperparameter specifying the segment length, divides a long time series into multiple equal-length time sequence segments, each considered as a local region. By setting different segment lengths, we obtain long sequences composed of equal-length segments representing multiple time scales.

Furthermore, the model incorporates the local attention mechanism, performing segment-based attention computations within each long sequence. This allows the model to explicitly extract local features from the temporal relationships of sub-sequence time series, capturing the temporal dependencies of multiple sub-sequence time series more effectively. The local attention mechanism facilitates interaction and integration of information within long sequences, enhancing the model’s expressive power. By applying the local attention mechanism to equal-length segments of various segment lengths constituting different long sequences, we efficiently acquire local features at different time scales.

Finally, all features are fused, and the prediction result is obtained through fully connected layers. Overall, the BiLSTM-MLAM model, by considering temporal information at different time scales and local dependencies comprehensively, enhances its accuracy in predicting future time series.

### 3.2. Bi-LSTM

Recurrent neural networks (RNNs) can capture dependencies within sequences and have shown good results when training on short sequences in the past. However, they suffer from issues such as exploding or vanishing gradients when dealing with long sequences due to having only one hidden state. Long short-term memory (LSTM) was invented by Hochreiter [44] to solve this issue. LSTM networks incorporate memory cells and gate mechanisms, allowing them to better capture and retain long-term dependencies within sequential data. The structure of the LSTM network unit is illustrated in Figure 2.

LSTM introduces an output based on the ordinary output $h_{t}$ of each unit, which is the memory cell $C_{t}$, and adds three gates: forget gate, input gate, and output gate. With the control of these three gates, it can regulate the flow of information, enabling LSTM to better handle and store long-term dependent information, thereby improving its memory capacity. The gating structure, hidden layer output, and cell state transfer process of the LSTM unit are shown in Equation (Equation 1) to Equation (Equation 6).
(1)$$f_{t}=\sigma \left(w_{f}\left[h_{t-1},x_{t}\right]+b_{f}\right)$$
(2)$$i_{t}=\sigma \left(w_{i}\left[h_{t-1},x_{t}\right]+b_{i}\right)$$
(3)$${\tilde{C}}_{t}=\tanh\left(w_{C}\left[h_{t-1},x_{t}\right]+b_{C}\right)$$
(4)$$C_{t}=f_{t}*C_{t-1}+i_{t}*{\tilde{C}}_{t}$$
(5)$$o_{t}=\sigma \left(w_{o}\left[h_{t-1},x_{t}\right]+b_{o}\right)$$
(6)$$h_{t}=o_{t}*\tanh\left(C_{t}\right)$$

The forget gate ($f_{t}$) decides what information to discard from the previous cell state. The input ($x_{t}$) provides current information, while $h_{t-1}$ represents the previous hidden state. The sigmoid activation function (σ) determines the forgetfulness probability. The input gate ($i_{t}$) selects new information to store, combining it with temporary memory (${\tilde{C}}_{t}$). $C_{t}$ is the current cell state, with $f_{t}C_{t-1}$ representing forgotten information and $i_{t}{\tilde{C}}_{t}$ denoting new input. The output gate ($h_{t}$) controls output to the hidden layer using sigmoid and tanh activations, resulting in the final output.

However, LSTM relies solely on the previous time step’s data and past information for predictions, potentially overlooking future contextual information. In order to address this shortcoming of LSTM networks, the suggested LSTM network in this research incorporates a bidirectional topology that allows it to process data over the full-time range. This enables the utilization of both preceding features and the inclusion of forthcoming information. The schematic representation of the Bi-LSTM concept is illustrated in Figure 3.

Bi-LSTM employs two distinct hidden layers: the forward hidden layer transmits information from the past to the future, while the backward hidden layer conveys information from the future to the past. In deep learning architectures, Bi-LSTM shows enhanced data representation capabilities compared to conventional LSTM. The output in Bi-LSTM is elucidated as follows:(7)$$h_{t}^{f}=\mathrm{LSTM}\left(x_{t},h_{t-1}^{f}\right)$$
(8)$$h_{t}^{b}=\mathrm{LSTM}\left(x_{t},h_{t-1}^{b}\right)$$
(9)$$y_{t}=W_{o}h_{t}+b_{o}$$

$W_{hy}^{f}$ represents the weights from the forward layer to the output layer, $W_{hy}^{b}$ represents the weights from the backward layer to the output layer, and $b_{o}$ is the bias vector of the output layer. $h_{t}$ is composed of integrating $h_{t}^{f}$ and $h_{t}^{b}$. At time *t*, Bi-LSTM simultaneously utilizes both past and future data, combining the information from both parts for learning.

### 3.3. Multi-Scale Patch Segmentation

Considering the strong local nature of time series, characterized by continuity between adjacent points, we will conduct segment-wise processing and maintain continuity within each segment during aggregation. Given a time series of length *t*, by setting patchsize equal to patchstep equal to *L*, where *L* is the length, patches are generated, dividing the time series into different segments. Let $X_{i}$ represent the i_th segment.
(10)$$X_{i}=X\left[i\cdot L_{\mathrm{seg}}:\left(i+1\right)\cdot L_{\mathrm{seg}}\right],1\leq i\leq \left\lfloor \frac{T}{L_{\mathrm{seg}}}\right\rfloor$$

$\left\lfloor .\right\rfloor$ represents the floor function, and yields the following:(11)$$X=\left[X_{1},X_{2},\ldots \right]$$

By configuring different patch sizes, sequences of varying lengths corresponding to different time scales can be obtained, as shown in Equation (Equation 12), allowing the capture of patterns at different temporal resolutions.
(12)$$\left(X,L_{{\mathrm{seg}}_{1}},L_{{\mathrm{seg}}_{2}},\ldots \right)=\left[X1,X2,\ldots \right]$$
where $L_{\mathrm{seg}1}$, $L_{\mathrm{seg}2}$ represent the segment lengths at different scales, and X1, X2 are sequences composed of segments from different time scales. $L_{\mathrm{seg}}$ determines the resolution in the calculation of the minimum unit involved in segment correlation. A larger $L_{\mathrm{seg}}$ implies capturing coarse-grained time dependencies in the time series, while a smaller $L_{\mathrm{seg}}$ can capture fine-grained dependencies. Through comprehensive learning, multi-granularity features are obtained, utilizing multiscale information to aid in predictions. Figure 4 provides examples of multiscale segmented sequences obtained with segment lengths of 2, 4, and 8, respectively.

### 3.4. Local Attention Mechanism

The attention mechanism focuses on key positions, reducing the weighting of non-key positions, and highlighting more relevant influencing factors, aiding the model in making accurate judgments. After segmenting the time series, the attention mechanism is applied on a segment-by-segment basis. For long sequences $X=\left[x_{1},x_{2},\ldots \right]$ at a specific time scale, the local attention mechanism module is applied to effectively capture the correlations between different segments. The structure of the local attention mechanism is illustrated in Figure 5.

For the input time series segment $x_{i}$, it is mapped to the query matrix $q_{i}$, key matrix $k_{i}$, and value matrix $v_{i}$ for that time segment through three learnable mapping matrices. The specific calculation process is as follows:(13)$$q_{i}=W^{q}x_{i}$$
(14)$$k_{i}=W^{k}x_{i}$$
(15)$$v_{i}=W^{v}x_{i}$$

The correlation between different time segments is as follows:(16)$$\alpha_{i,j}=q_{i}⊙k_{j}$$
where ⊙ denotes element-wise multiplication, and the correlation between time segments is measured by calculating the element-wise multiplication of the query matrix *q* and the key matrix *k*. The correlations between multiple time segments are obtained by computing the query matrix and key matrix, the softmax function is applied for normalization. The calculation formula is as follows:(17)$${\tilde{x}}_{i}=\sum_{j=1}^{t}\alpha_{1,j}^{\prime }v_{j}$$

Concatenate the outputs ${\tilde{x}}_{i}t$ of multiple time segments along the time dimension. The original sequence *X* is transformed into a time series $\tilde{X}=\left\{{\tilde{x}}_{1},{\tilde{x}}_{2},\ldots ,{\tilde{x}}_{n}\right\}$, representing the learned features through the local self-attention mechanism module. Here, *n* denotes the number of segments into which the long sequence *X* is divided.

For different time-scale segment sequences X1,X2…, apply the local attention mechanism module to obtain $\tilde{X}1,\tilde{X}2\ldots$; concatenate these results and feed them into a fully connected layer to obtain the final prediction result.

## 4. Experiments and Simulation Results

In this section, we present a comparative analysis of our approach against six baseline methods using three public datasets, showcasing its superior performance in time series prediction. Furthermore, an ablation experiment is performed to affirm the efficacy of each component in the proposed method. Lastly, we explore the influence of diverse patch size parameter configurations and their merging on prediction performance.

### 4.1. Data Description

This study validates and compares results using three publicly available datasets: the C-MAPSS dataset, the LTE dataset, and the load dataset. Detailed information for each dataset is summarized in Table 1.

C-MAPSS: This dataset is designed for predicting the remaining lifespan of aircraft engines and is publicly released by the National Aeronautics and Space Administration (NASA) of the United States. The dataset is divided into four subsets, namely FD001 to FD004, each comprising both training and testing datasets. Each training and testing dataset includes engine ID, operational cycles, and three operational conditions (flight altitude, Mach number, and throttle resolver angle), along with 21 additional sensor values, totaling 26 columns. Since previous studies on time series prediction have predominantly concentrated on the FD001 subset, we chose to employ the FD001 dataset in our experiments for ease of comparison. Specifically, our training set includes data from 100 engines, while the testing set also encompasses data from 100 engines. The training set provides data on turbofan engine samples from operation to failure, whereas the testing set comprises turbofan engine samples under identical operating conditions, covering only the front half of the engine’s operational cycles, and includes the true remaining useful life (RUL) values for each sample. The C-MAPSS dataset is available at https://ti.arc.nasa.gov/tech/dash/groups/pcoe/prognostic-data-repository/, accessed on 18 June 2023.

To process the C-MAPSS dataset, we followed these steps:Data loading: We imported the dataset from the provided source, incorporating both the training and testing datasets for the FD001 subset.Feature selection:We identified relevant features such as engine ID, operational cycles, operational conditions, and sensor values.Data preprocessing:−Handling missing values: We checked for any missing values and applied appropriate techniques such as imputation or removal.−Standardization: We ensured that numerical features were standardized to have a mean of 0 and a standard deviation of 1.Data splitting: We segmented the dataset into training and testing sets, ensuring a representative distribution of engine samples.Target generation: We derived the target variable, remaining useful life (RUL), for the training set by calculating the remaining operational cycles until failure.Final Dataset Preparation: We formatted the datasets for model training and evaluation, ensuring proper alignment and compatibility.

LTE: This dataset, sourced from Kaggle, is designed for predicting LTE 4G network traffic. The dataset encompasses traffic data from 57 base stations in a specific region, covering the period from 23 October 2017 to 22 October 2018. Hourly sampling was conducted, resulting in 24 samples per day. For our experiments, we focused on a three-month subset of the data to assess the proposed model. The LTE dataset can be accessed at https://www.kaggle.com/naebolo/predict-traffic-of-lte-network, accessed on 28 June 2022.

Load dataset: This dataset is intended for load prediction and originates from historical data on electrical, cooling, and heating loads at the Tempe campus, provided by the Campus Metabolism system at Arizona State University. The data spans from 1 January 2019, at 0:00, to 31 July 2023, at 24:00, with a sampling interval of 1 h. The load dataset is available at https://cm.asu.edu, accessed on 8 June 2022.

The LTE and load datasets underwent similar data processing steps to the C-MAPSS dataset, encompassing data loading, feature selection, data preprocessing, data splitting, target generation, and other relevant procedures.

### 4.2. Model Evaluation Criteria

The root mean square error (RMSE) and mean absolute error (MAE) serve as evaluation criteria. These metrics are employed to quantify the disparity between the observed and predicted data, and are described as follows:(18)$$E_{\mathrm{RMSE}}=\sqrt{\frac{1}{n}\sum_{i=1}^{n}{\left({\hat{y}}_{t}-y_{t}\right)}^{2}}$$
(19)$$E_{\mathrm{MAE}}=\frac{1}{n}\sum_{i=1}^{n}\left|{\hat{y}}_{t}-y_{t}\right|$$
where $y_{t}$ represents the observed responses, ${\hat{y}}_{t}$ represents the estimated responses, and n is the total number of observations.

### 4.3. Experiments and Discussions

#### 4.3.1. Model Comparison

In order to prove that the suggested BiLSTM-MLAM approach is preferable, we carried out a short-term prediction experiment comparing BiLSTM-MLAM with six other methods. The C-MAPSS dataset is designed to predict the RUL of aircraft engines, focusing primarily on the engine’s life status, to forecast the life status within the next operating cycle. The LTE and load datasets are collected at an hourly frequency and used to predict the network traffic in a specific region and the load in the Arizona State University Tempe campus for the next time interval. In this experiment, the models compared are as follows:SVR: In time series regression, SVR can adapt flexibly to different data distributions by selecting the optimal kernel function and relevant parameters, thereby providing accurate regression results.RNN: Compared to traditional feedforward neural networks, RNN introduces recurrent connections, enabling it to model sequential information and capture temporal dependencies within sequences.LSTM: This is a variant of RNN designed to overcome the vanishing gradient problem in traditional RNNs. It excels at handling long sequences and capturing long-term dependencies.GRU: Compared to LSTM, GRU has a more concise structure, with one fewer gate units, resulting in fewer parameters and easier convergence.Bi-LSTM: Bi-LSTM enhances traditional LSTM networks by analyzing input sequences bidirectionally. This enables the network to capture dependencies from both the past and future in the sequence, improving its understanding of temporal relationships in time series data.Encoder–decoder: This is a sequence-to-sequence model widely used in tasks such as machine translation. The model processes the source sequence step by step through an encoder, mapping it to a vector of fixed length.

We employed a grid search method to adjust the hyperparameters of each model. For SVR, the final decision was to use a linear kernel, with regularization parameters and gamma set to their default values. For baseline models, including RNN, LSTM, GRU, Bi-LSTM, and encoder–decoder, we conducted a grid search over the sizes of recurrent and dense layers, considering 16, 32, 64, 128, 256. We chose the patch size from 2, 3, 4 for BiLSTM-MLAM, taking efficiency and performance into consideration.

The prediction results of all approaches on three datasets are summarized in Table 2, where the best results for each dataset are bolded. It can be observed that, across all datasets, the proposed BiLSTM-MLAM outperforms other methods in both metrics. Specifically, BiLSTM-MLAM shows a 6.66% and 11.50% improvement in RMSE and MAE, respectively, compared to most comparative methods (encoder–decoder) in predicting the remaining life of aircraft engines. It demonstrates a 12.77% and 3.06% enhancement in predicting base station network traffic and a 17.96% and 30.39% improvement in load prediction. Figure 6, Figure 7 and Figure 8 visualize the prediction results of BiLSTM-MLAM on three datasets It can be seen that BiLSTM-MLAM fits the actual values well. These results strongly attest to the excellence of BiLSTM-MLAM in time series forecasting.

#### 4.3.2. Ablation Study

To illustrate the usefulness of the suggested approach and investigate how each element affects prediction performance, we used the C-MAPSS dataset for an ablation investigation. By changing or eliminating individual components one at a time in BiLSTM-MLAM, we obtained three variants:Bi-LSTM: Without performing multi-scale segmentation fusion and local attention mechanism processing.BiLSTM-AM: Utilizing the attention mechanism but without the multi-scale fusion mechanism.LSTM-MLAM: Using a single-layer LSTM.

Table 3 presents the experimental results, from which the following conclusions can be drawn:BiLSTM-MLAM achieved the best results across all metrics;BiLSTM-AM outperformed Bi-LSTM, demonstrating that the attention mechanism effectively improves prediction performance;Removing multi-scale patch segmentation significantly reduced the model’s prediction accuracy, highlighting the effectiveness of multi-scale fusion;Compared to LSTM-MLAM, BiLSTM-MLAM showed significant improvements, revealing the crucial role of the bidirectional structure.

In conclusion, each component of the BiLSTM-MLAM model effectively enhances prediction performance.

#### 4.3.3. Multi-Scale Feature Fusion

In this section, we delve into the impacts of different segment length parameters on prediction performance within the multi-scale feature fusion mechanism. Specifically, we selected the C-MAPSS dataset and fixed the input data length (T) at 24.

The patch size parameter represents the length of segments in a long time series, influencing the model’s extraction of features across different time scales. In this experiment, we set the patch size to 2, 3, 4, representing various scales of feature extraction. In order to confirm that the multi-scale feature fusion technique works as intended, we conducted experiments by combining features extracted from two or three scales. The summarized experimental results are presented in Table 4.

Observing Table 4, we note the significant variations in prediction errors with different patch sizes, attributed to the inconsistency in feature granularity. As the variety of patch sizes in the fusion increases, the model’s prediction error gradually decreases. This is because the multi-scale fusion extracts more comprehensive features, enabling better capturing of changes across different scales in the time series.

## 5. Conclusions

This paper presents a multi-scale time series prediction model named BiLSTM-MLAM, which integrates bidirectional long short-term memory (Bi-LSTM) and local attention mechanism (LAM). By capturing sequence information at different time scales and enhancing feature extraction through the local attention mechanism, BiLSTM-MLAM demonstrates exceptional performance in time series prediction tasks. Experimental results indicate that, compared to alternative methods, BiLSTM-MLAM achieves superior results across multiple metrics, validating its outstanding capabilities on diverse datasets such as C-MAPSS, LTE, and Load. Additionally, the ablation experiments emphasize the effectiveness of the local attention mechanism and multi-scale fusion strategy, as well as the crucial role of the bidirectional LSTM architecture in capturing dependencies in time series data. Segment length parameter tuning experiments further confirm the significant impact of multi-scale fusion on improving prediction accuracy, particularly by optimizing combinations of different segment lengths to effectively enhance model performance. For the remaining useful life prediction of aircraft engines, BiLSTM-MLAM effectively captures the time series characteristics of engine operational data, enabling more accurate life predictions. This aids in optimizing maintenance and operational planning, thereby enhancing safety and economic efficiency. In LTE network traffic prediction, the model can handle large and complex traffic data, providing high-precision traffic forecasts. This is of significant practical value for network operators in resource allocation, congestion management, and ensuring service quality. In energy load forecasting, BiLSTM-MLAM can help energy management systems more accurately predict load demand, and optimize energy distribution and scheduling, thus improving energy utilization efficiency and reducing operational costs.

In the future, we plan to conduct in-depth research building upon the foundation of our proposed model. The focus will primarily be on methods and strategies related to feature extraction and multi-scale feature fusion. Our intention is to explore diverse approaches for extracting features at different granularities and optimizing the fusion of these features. Specifically, we aim to acquire temporal information from multiple perspectives, which may involve finer time partitioning and more flexible feature selection. By delving deeper into the intrinsic patterns of time series data, we aim to devise a more precise predictive model, further enhancing the accuracy of time series forecasting. 

## Figures and Tables

**Figure 1 sensors-24-03962-f001:**
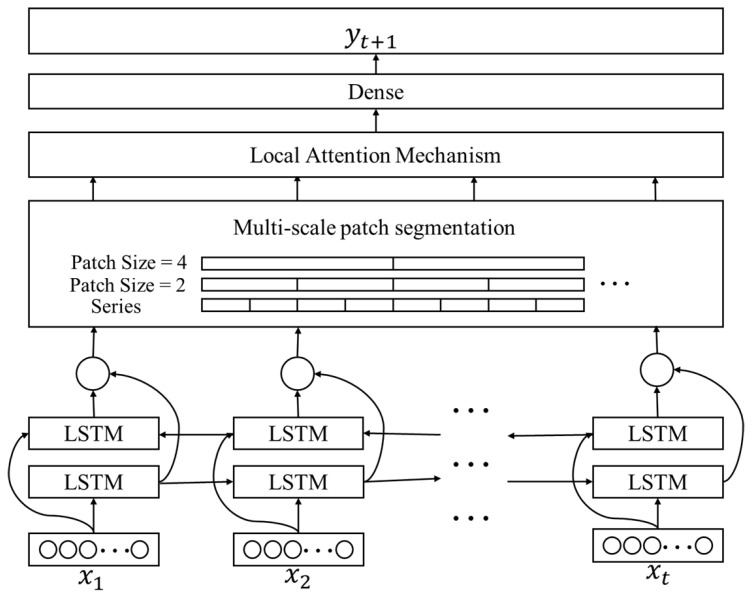
Overall framework of the model.

**Figure 2 sensors-24-03962-f002:**
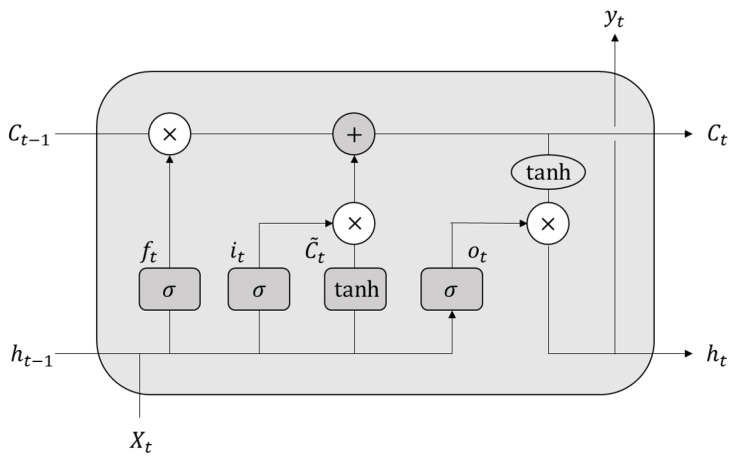
The basic structure of the LSTM neural network.

**Figure 3 sensors-24-03962-f003:**
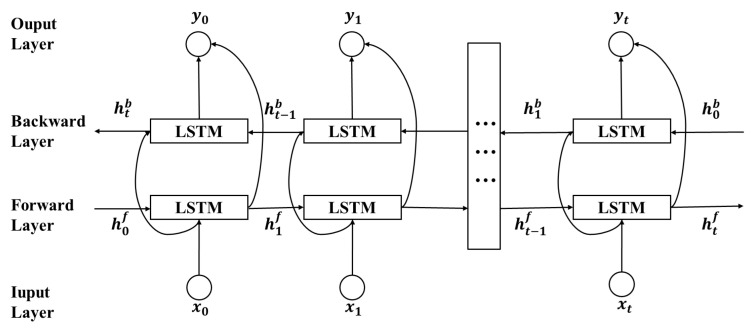
The Bi-LSTM network structure.

**Figure 4 sensors-24-03962-f004:**
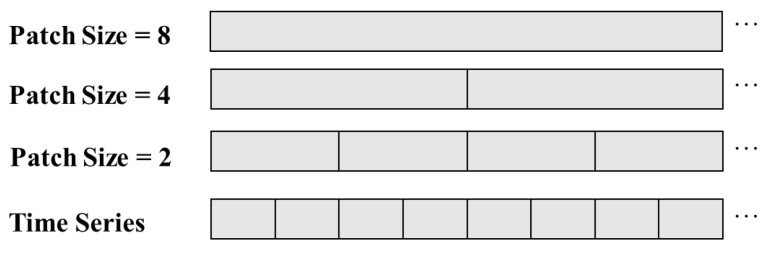
Multi-scale patch segmentation.

**Figure 5 sensors-24-03962-f005:**
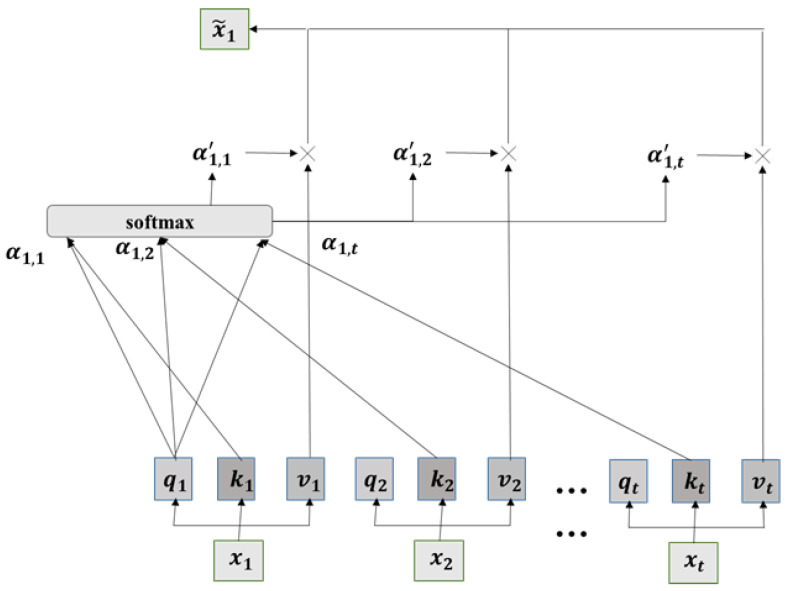
The structure and computational process of the LAM.

**Figure 6 sensors-24-03962-f006:**
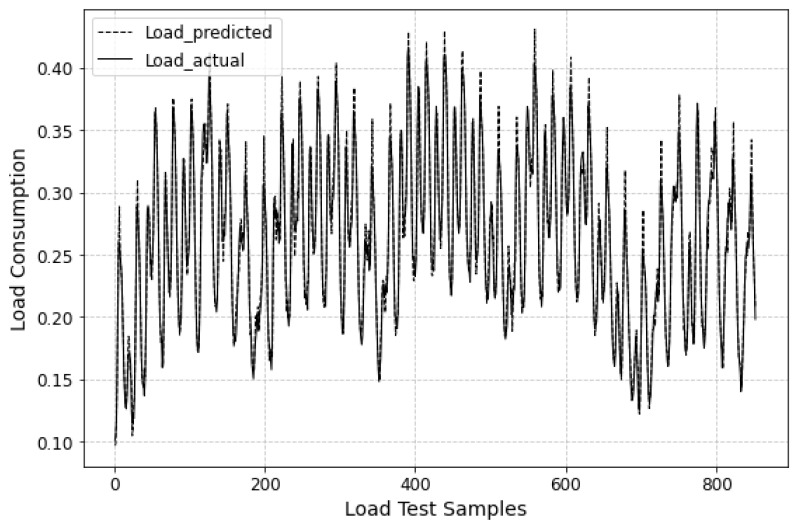
Prediction results of BiLSTM-MLAM on C-MAPSS.

**Figure 7 sensors-24-03962-f007:**
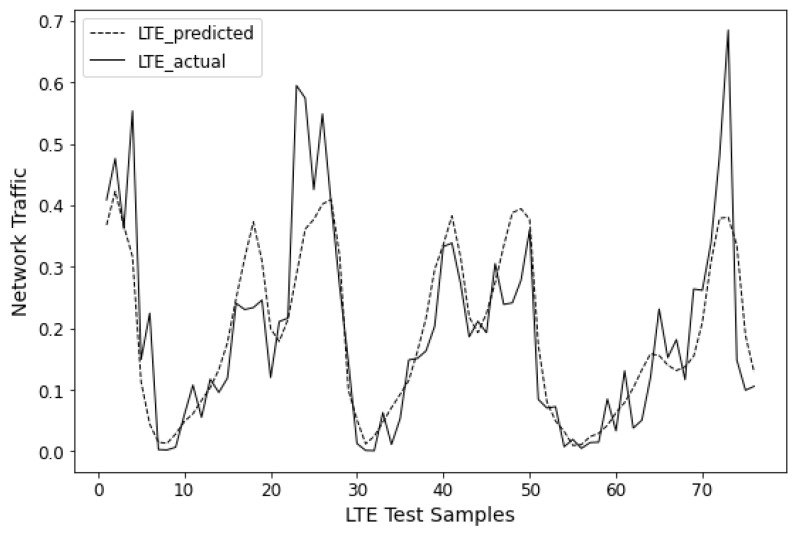
Prediction results of BiLSTM-MLAM on LTE.

**Figure 8 sensors-24-03962-f008:**
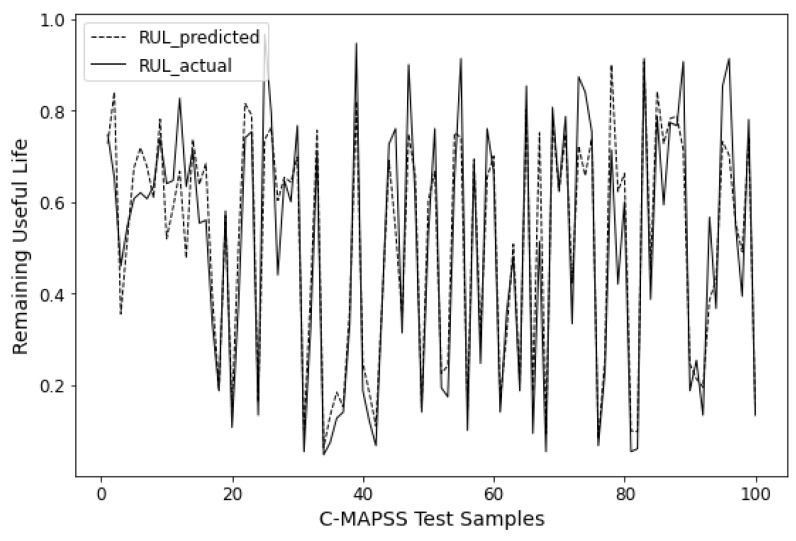
Prediction results of BiLSTM-MLAM on Load.

**Table 1 sensors-24-03962-t001:** Dataset details.

Datasets	Total Size	Sample Rate
C-MAPSS	17,731	1 cycle
LTE	26,214	1 h
Load	17,520	1 h

**Table 2 sensors-24-03962-t002:** Prediction results of all models.

Models	C-MAPSS		LTE		Load	
	RMSE	MAE	RMSE	MAE	RMSE	MAE
SVR	0.1212	0.0950	0.1132	0.0766	0.0141	0.0112
RNN	0.1049	0.0763	0.1003	0.0685	0.0124	0.0091
LSTM	0.1007	0.0761	0.0983	0.0707	0.0111	0.0086
GRU	0.1278	0.0966	0.0956	0.0657	0.0116	0.0088
Bi-LSTM	0.1028	0.0756	0.0873	0.0633	0.0108	0.0079
Encoder–Decoder	0.1005	0.0817	0.0971	0.0620	0.0128	0.0102
BiLSTM-MLAM	**0.0938**	**0.0723**	**0.0847**	**0.0601**	**0.0105**	**0.0071**

Note: The bold numbers indicate the best results for each dataset.

**Table 3 sensors-24-03962-t003:** Results of ablation study.

	C-MAPSS	
Models	RMSE	MAE
Bi-LSTM	0.1028	0.0756
BiLSTM-AM	0.0998	0.0741
LSTM-MLAM	0.0984	0.0788
BiLSTM-MLAM	0.0938	0.0723

**Table 4 sensors-24-03962-t004:** Prediction results of BiLSTM-MLAM with different patch sizes on the C-MAPSS dataset.

	C-MAPSS	
Patch Configuration	RMSE	MAE
2	0.0984	0.0788
3	0.0981	0.0725
4	0.1038	0.0731
2&3	0.0977	0.0724
2&3&4	0.0938	0.0723

## Data Availability

The data utilized in this paper were sourced from publicly available resources, including the C-MAPSS dataset from https://ti.arc.nasa.gov/tech/dash/groups/pcoe/prognostic-data-repository/, accessed on 18 June 2023, the LTE dataset from https://www.kaggle.com/naebolo/predict-traffic-of-lte-network, accessed on 28 June 2022, and the load dataset from https://cm.asu.edu, accessed on 8 June 2022.

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
