# Peer review of "BiLSTM-MLAM: A Multi-Scale Time Series Prediction Model for Sensor Data Based on Bi-LSTM and Local Attention Mechanisms"

_sensors, 2024, doi:10.3390/s24123962_

Round 1
Reviewer 1 Report
Comments and Suggestions for Authors
This paper introduces an innovative multi-scale time series prediction model utilizing Bidirectional Long Short-Term Memory (Bi-LSTM) to capture information from both forward and backward directions in time series data. It demonstrates outstanding performance in time series prediction tasks by effectively capturing sequence information across various time scales. However, revisions are necessary before publication. Therefore, I recommend Major Revision. The specific review comments are outlined below:
-
The current analysis of the Prediction Model Based on Bi-LSTM and Local Attention Mechanism is deemed insufficient. In the field of prediction, several aspects merit further research, such as methods for recovering missing measurement data, advanced prediction models, and optimization techniques like swarm intelligence. It is advised that the authors incorporate appropriate discussions in the literature review section.
-
The theory of Bi-LSTM and Local Attention Mechanism in the Methods section should be presented in a simplified manner. The authors have mainly applied this method to prediction with few enhancements. Any improvements made by the authors should be clearly delineated.
-
While the prediction results in Figures 6 and 8 are satisfactory, Figure 7 shows poor performance. The authors are urged to provide an explanation for this inconsistency.
-
The authors should furnish detailed introductions for the three datasets or specify their sources in the paper.
-
If feasible, the authors are encouraged to supplement the discussion on the practical application of the proposed method or provide explanations in the Conclusion section regarding its applicability in real-world scenarios.
Reviewer 2 Report
Comments and Suggestions for Authors
This paper introduces a novel time series prediction model based on Bi-LSTM (Bidirectional Long Short-Term Memory) and Local Attention Mechanism.
It is shown that for some time series (with public access) the new proposed model (BiLSTM-MLAM) gives better prediction. However, the results are not presented in the best form for the Sensors journal. Since the subject matter of the paper is not appropriate for the journal.
I recommend submitting this paper to other journals on the subject of prediction, statistics of artificial intelligence, etc. The peer review in these journals will be of better quality on the features of the proposed model.
In another case, the article should be seriously revised to be consistent with the theme of the issue - "Sensor Networks". In this case you need to give examples describing sensor networks, measurement and other problems that can be solved with the proposed approach.
In your article the term sensor appears only 1 time, as an example dataset C-MAPSS. This is not enough.
There is also an unfortunate typo: the last 2 sentences in the conclusion are missing (lines 416-419).
The paper can be published after major revision. But I recommend submitting the paper in this form to another journal rather than doing a complete revision of the paper for this journal and not wasting time.
Reviewer 3 Report
Comments and Suggestions for Authors
No
Reviewer 4 Report
Comments and Suggestions for Authors
The paper introduces a novel multi-scale time series forecasting model BiLSTM-MLAM (Bi-LSTM with Local Attention Mechanism and Multi-scale Patch Segmentation). This model combines the advantages of bidirectional LSTMs (Bi-LSTMs) to capture information from both the past and future, a local attention mechanism for effective feature extraction, and multi-scale feature fusion.
The model begins by passing the input time series through a Bi-LSTM to extract sequential information. The multi-scale patch segmentation module then divides the long time series into segments of different lengths representing various time scales. The local attention mechanism is applied to these constituent segments to accurately extract local features from the temporal dependencies within the segments. Fusing the extracted multi-scale features allows the model to capture patterns at different time resolutions.
In experiments on the C-MAPSS, LTE, and Load datasets, the proposed BiLSTM-MLAM outperformed six baseline models on the RMSE and MAE metrics for time series forecasting tasks. Ablation analysis confirmed the effectiveness of each component in the model. Experiments with different segment lengths showed that combining multiple scales reduced forecasting errors, validating the benefits of the multi-scale strategy.
Through extensive experiments and analysis, BiLSTM-MLAM demonstrates its outstanding performance as a highly effective time series forecasting model, capable of extracting informative features across different time scales and capturing intricate dependencies in the data, exhibiting superior forecasting accuracy across diverse datasets.
Overall, this paper presents a novel and promising approach to multi-scale time series forecasting through the proposed BiLSTM-MLAM model. The authors have demonstrated a comprehensive understanding of the challenges involved in time series forecasting and have proposed an innovative solution that effectively addresses these challenges. The experimental results, backed by thorough evaluations across multiple datasets and comparative analyses with baseline methods, provide strong evidence of the model's superior performance and forecasting accuracy. Despite the potential for further improvements, such as additional real-world dataset evaluations, interpretability analysis, and computational complexity considerations, the paper's contributions are significant and represent a valuable addition to the field of time series forecasting. Therefore, I believe that this paper could be accepted for publication.
Round 2
Reviewer 1 Report
Comments and Suggestions for Authors
Accept.
Reviewer 2 Report
Comments and Suggestions for Authors
In the corrected version of the article the authors justified the relevance of the conducted research to the topic of the journal Sensors. The article is formally suitable for publication in this journal .
The paper can be published in present form.
Reviewer 3 Report
Comments and Suggestions for Authors
The authors have replied my concerns.